# Non-Invasive Driver Drowsiness Detection System

**DOI:** 10.3390/s21144833

**Published:** 2021-07-15

**Authors:** Hafeez Ur Rehman Siddiqui, Adil Ali Saleem, Robert Brown, Bahattin Bademci, Ernesto Lee, Furqan Rustam, Sandra Dudley

**Affiliations:** 1Faculty of Computer Science and Information Technology, Khwaja Fareed University of Engineering and Information Technology, Rahim Yar Khan 64200, Pakistan; adilalisaleem@gmail.com (A.A.S.); furqan.rustam1@gmail.com (F.R.); 2School of Engineering, London South Bank University, London SE1 0AA, UK; brownr16@lsbu.ac.uk (R.B.); bademcb3@lsbu.ac.uk (B.B.); dudleyms@lsbu.ac.uk (S.D.); 3Department of Computer Science, Broward College, Broward County, FL 33332, USA; 4Department of Business Administation, Baker College, Owosso, MI 48867, USA

**Keywords:** drowsiness detection, respiration rate, physiological signals, machine learning, ultra-wideband

## Abstract

Drowsiness when in command of a vehicle leads to a decline in cognitive performance that affects driver behavior, potentially causing accidents. Drowsiness-related road accidents lead to severe trauma, economic consequences, impact on others, physical injury and/or even death. Real-time and accurate driver drowsiness detection and warnings systems are necessary schemes to reduce tiredness-related driving accident rates. The research presented here aims at the classification of drowsy and non-drowsy driver states based on respiration rate detection by non-invasive, non-touch, impulsive radio ultra-wideband (IR-UWB) radar. Chest movements of 40 subjects were acquired for 5 m using a lab-placed IR-UWB radar system, and respiration per minute was extracted from the resulting signals. A structured dataset was obtained comprising respiration per minute, age and label (drowsy/non-drowsy). Different machine learning models, namely, Support Vector Machine, Decision Tree, Logistic regression, Gradient Boosting Machine, Extra Tree Classifier and Multilayer Perceptron were trained on the dataset, amongst which the Support Vector Machine shows the best accuracy of 87%. This research provides a ground truth for verification and assessment of UWB to be used effectively for driver drowsiness detection based on respiration.

## 1. Introduction

Drowsiness is a state of tiredness that results in heavy eyelids, daydreaming, rubbing of eyes, loss of focus and yawning. Drowsiness is one of the main causes of fatal crashes. According to a recent investigation, 1 million people have died in road accidents [1], 30% of which have been caused by driver fatigue or drowsiness [2]. The chances of having a crash are three times higher if the driver is fatigued [3]. According to the American Automobile Association (AAA), there were 328,000 drowsy driving crashes annually, costing 109 billion USD to society, excluding property damage [3]. During a 4 or 5 s bout of driver inattention, a vehicle can almost cover the length of a football field before stopping [3]. Reports reveal that night-shift male workers and people with sleep apnea syndrome are at the highest risk of becoming drowsy during driving [4]. Research investigations have been published that proposed methods to counteract or alert drivers about potential signs of drowsiness [5,6,7,8,9,10].

Drowsiness detection systems are divided into three main categories: vehicle dynamics, physiological signals and driver facial characteristic recognition [5]. Performance of the vehicle dynamics-based drowsiness detection systems is low due to the impact of unpredictable factors such as road geometry, slow processing speed and traffic [5], and additionally, head movement. Yawning and blinking extracted from the driver facial images have shown promising results in controlled or virtual environments [5]. In a real environment, the performance of these systems decreases due to the impact of factors such as light variations, skin color and temperature, etc. [5].

On the other hand, systems using physiological signals gave accurate results that make them a reliable approach to use in the real environment [5]. Physiological signals such as electroencephalography (EEG) [11,12,13], electrooculography (EOG) [13,14,15], respiration rate [16], electrocardiography (ECG) [17,18,19] and electromyography (EMG) [20,21] signals are frequently highlighted in driver drowsiness detection systems. However, most of these signals are acquired using invasive sensors, making them hard to integrate or employ in real environments. The respiration system undergoes significant changes from wakefulness to sleep and varies based on different physiological conditions. However, the respiration rate is the least-measured vital sign due to uncomfortable nature of respiration rate acquiring methods [22]. Breathing during sleep is affected by a reduction of muscular tone and the alteration of chemical and non-chemical responses [23]. Respiration rate usually decreases before the driver falls asleep [24,25,26].

The research presented in this manuscript aims towards the detection of driver drowsiness by non-invasively acquiring chest movement with impulse radio ultra-wideband (IR-UWB) radar. Impulse radio ultra-wideband (IR-UWB) radar is an evolving technology. It was first used by the US army in 1973 and commercialized by the Time domain and Xtreme Spectrum companies in the late 1990s [27,28]. USA-based Federal Communication Commission (FCC) allocated a bandwidth of 7.5 GHz for UWB signals [27]. This bandwidth covers a frequency of 3.1–10.6 GHz, and a signal is considered as UWB if that has a bandwidth of 500 MHz [27]. UWB signals have high data rates and low power transmission levels, producing high bandwidth signals due to the transmission of very short duration pulses. IR-UWB radar transmits up to 10 million nano-pulses per second to gather valuable information that enables the detection and monitoring of micro-movements and vibrations such as breathing and heartbeats. The IR-UWB radar does not raise any privacy issues and is not affected by environmental factors as it has no light or skin-color dependencies. Radar does not have any harmful effect on the human body as the emission power of the IR-UWB radar is extremely low (limited to −41.3 dBm/MHz) [29,30,31]. Additionally, this system does not suffer in the presence of Wi-Fi and mobile phone signals. IR-UWB radar has advantages over other existing tools due to its non-intrusive, non-tackling capabilities and its potential to penetrate through different materials or obstacles [32,33]. Various investigations have been performed on UWB-based wireless sensing devices to detect vital signs for health care applications [33,34,35,36]. To the authors’ best knowledge, there is no current system to detect drowsiness using data-analysis techniques based on UWB. In this method, the respiration rate is estimated from the acquired chest movement, and a dataset is maintained along with age and labels of drowsy/non-drowsy. Different ML algorithms are trained and evaluated on the dataset.

The rest of this article is organized as follows. Related work is discussed in Section 2. Section 3 presents the experimental methodology. Results and Discussion are provided in Section 4 and Section 5, respectively.

## 2. Related Work

In recent years, several respiration-monitoring methods for drowsiness detection have been introduced. A driver drowsiness detection system was proposed by [37] using respiration signals acquired by a pulse oximeter. The respiration signal is obtained in pre- and post-driving states of 150 professional male drivers. During the experiments, classification accuracy using Daubechies wavelet at decomposition level 3 (CA3) and classification accuracy using DMeyer wavelet at decomposition level 4 (CA4) showed 100% accuracy using MIN, MAX, Mean and Mode features.

In [38], a non-invasive method was presented that acquires breathing rate using two high-dynamic cameras, PAC16 and FRCAM, to detect driver drowsiness. Five males with ages ranging between 28 and 38 years took part in the data collection process. The subjects’ respiration signals in normal conditions and when sleep deprived (24 hours without sleep) were acquired during a 1 h and 40 m driving simulation. The experiment was conducted in a controlled environment to reduce noise in the data collection process. The videos were recorded using cameras and were converted to frames. Histogram equalization was applied to increase global contrast to control lightning conditions in an outdoor environment. Noise filtering and image stabilization were applied to counteract the effects of motion during vehicle driving. Frame-differentiation-based techniques were applied to quantify motion level, and the image was subsequently segmented in regions where motion is detected. Non-periodic signals were discarded after analyzing the motion signals of each image segment. A short-term Fourier transform was applied over the motion signals to calculate the respiration rate.

An optical imaging technique is used to acquire physiological signal brainwave, cardiac and respiration pulses for fatigue detection in [39]. Driver’s facial images are captured using an Infrared (IR) camera placed on the dashboard. Physiological signals, heart rate (HR), heart rate variability (HRV) and respiration are also acquired from captured images using photoplethysmography (PPG) as acquired by [40]. PPG is an optically obtained plethysmogram that can be used to detect blood volume changes in the microvascular tissue. Viola–Jones algorithm was used to detect the face region [41]. PPG extracts blood volume pulse (BVP) from a sequence of facial images [42]. The HRV was acquired from BVP by applying a Butterworth filter of window size 15 and bandwidth of 0.75–4 Hz. Respiration rate was acquired from the center of frequency of HF that was in between 0.15–0.4 Hz of HRV power spectrum density (PSD). These extracted facial and physiological features are subsequently fed into multiclass SVM [39].

Facial thermal imaging respiration analysis was proposed in [43] to detect driver drowsiness. Respiration causes temperature changes under the nostrils which was detected by thermal imaging. The respiration region was detected by the geometrical features of the face, and then a target tracker was used to track this region in subsequent frames. In this study, the respiration region was detected in the first few seconds of the thermal image sequence. To obtain an accurate respiration region, the driver’s head should not make any rapid movements during the first 5 s of the thermal imaging process. Noise was eliminated by applying a fourth-order Butterworth low-pass filter with a cut-off frequency of 0.6 Hz. Two machine learning models, SVM and KNN, were used with the fusion of all extracted features; SVM showed 90% accuracy, which is better than the accuracy (83%) showed by KNN.

Multiple Physiological signals, including oximetry pulse (OP), skin conductance (SC) and respiration signals, were acquired for fatigue detection by tagging respective sensors of the Nexus-10 [44]. Ten random drivers of taxies, lorries, luxury buses and trucks were selected for this study. The physiological signals of those drivers were recorded at a sampling rate of 256 Hz for 3–5 m before driving in the morning and after covering a distance of 500–600 km in a day. A median filter with a window length of 200 and a bandpass filter with a cut-off frequency of 200 Hz were applied to remove the baseline drift and noise, respectively. Hilbert–Huang transform (HHT) was used to remove ECG and EEG signals from the recorded signals. A Linear combination of the number of intrinsic mode functions (IMF) was acquired by decomposing the processed signals. Six features, namely, mean of the signals, maximum of the signal, standard deviation of the signals and mean, maximum and standard deviation of frequency, are extracted after dividing the signal into half-second frames. A scatter plot shows the inseparable overlapping of some features, and classification is performed by random forest (RF). The dataset is divided into 67% training and 33% testing data. RF combined with HHT gives an accuracy of 99% which is higher than MLP (93%) and SVC (70%).

A heart and respiration rate acquired using a safety belt for driver state recognition is presented by [45]. A textile cover for the seat belt comprising of an optical sensor and magnetic induction (MI) system is validated using ECGs and a piezoelectric sensor. It is observed during experiments that the proposed system gives better monitoring of the respiration rate but produces high-frequency signal noise that makes monitoring of heart rate difficult during inspiration.

The system proposed by [46] employed respiration rates derived from ECG signals for drowsiness detection. A portable fingertip pulse wave sensor placed on the left index finger was used to collect the acceleration pulse wave, which was very similar to the heart rate (can be considered as a heart rate signal). A non-intrusive driver fatigue detection was proposed using Continuous-Wave (CW) Doppler radar in [47]. CW radar placed on the car dashboard was used to acquire driver respiration and heart rate.** The collected heart and respiration rate signals were segmented into 60-second frames. A total of 240 data points were collected, and seven features in the frequency and time domains are extracted. A decision tree with an accuracy of 82.5% was used for classification purposes because it can easily process non-linear characteristics between values [48].

A driver drowsiness detection system based on respiration rate acquired using an inductive plethysmography belt is proposed in [49]. A system using HRV derived respiration measures to detect driver drowsiness using a wearable ECG device (Polar H7) is presented in [26]. RR-intervals (RRI) data with constant time intervals at the sampling rate of 0.5 s is acquired by performing Cubic interpolation. High frequency (HF), low frequency (LF), LFHF ratio and VLF power are used in predicting a drowsy state of the driver. Random forest (RF), KNN and SVM, are used to verify the usefulness of the drowsiness detection. SVM shows better accuracy among these three models.

According to the literature, the the vast majority of systems developed over the recent and past years employing physiological signals to identify driver drowsiness used invasive or on-body sensors. Invasive sensors can be utilized in virtual or controlled situations but cannot be employed in the actual environment since they cause driver distraction or discomfort while driving and consequently collect unrealistic data. As a result, a non-invasive, non-camera-based method for gathering physiological data that can be used in the real environment to detect drowsiness is required.

## 3. Materials and Methods

Real-time data collection and preprocessing is a key aspect in the development of any system that promotes safety and warning attributes. During the experimental process of data collection, the X4m300 ultra-wideband (UWB) radar (NOVELDA, Oslo, Norway) shown in Figure 1 was used. The UWB radar operates on the X4 system on chip (SoC) at the unlicensed center frequencies of 7.29 GHz or 8.748 GHz operating at bandwidth 1.4/1.5 GHz(−10 dB), with a configurable frame size [50]. UWB radar has excellent resolution range due to its nanosecond pulse transmission [51].

The sensor is powered by XeThru X4 UWB chip. The detection zone is adjustable up to 9.4 m with a detection time of 1.5 to 3.5 s [52]. Specific Absorption Rate (SAR) radiated by the radar is well below the limit value established by International Commission on Non-Ionizing Radiation Protection (ICNIRP) for the general population both for the SAR as averaged over the whole body and over 10 g [53].

Maintaining the boards default settings, the built-in firmware generates baseband signal covering distance of 9.4 m starting from 0.18 m. This distance is divided into 181 bins with bin length of 0.0514 m. Considering driver sitting position from dashboard, the effective range in the case of the presented manuscript is taken from within 0.2–1.6 m. The corresponding bins to this effective range are starting from second bin at 0.282 m to 28th bin at 1.569 m. The total number of bins in effective range is 27, as shown in Figure 2.

A Frame Counter is supplied with the baseband data. The Frame Counter increases by 1 for each radar frame that is output from the X4 UWB radar SoC. The frame counter size is 2^32. The frame counter wraps to 0 when it reaches the maximum, and a reset of the X4 UWB radar SoC or a power toggle of the sensor module will reset the frame counter [54].

The inhaling process extends the upper body, and the distance between the radar and human chest body decreases accordingly, while exhaling enacts the opposite. The energy intensity (amplitude) of electromagnetic wave decreases if it travels longer distance. When the distance between radar and the body is shorter, the received signal has higher energy (amplitude) as compared to the radar signals that travel over a longer distance. This phenomenon is reflected in the plot shown in Figure 2 by increasing (inhaling) and decreasing (exhaling) in amplitude at a specific distance corresponding to the target location with respect to the radar. These amplitude values of the received signal are stored in a matrix of 6000 rows and 27 columns. The number of rows depends on the time for which the radar scans, in this case, presented 6000 rows for 5 min scan and columns/bins represent distance with column/bin length is 0.0514 m.

Let the whole curve f(x) shown in Figure 2 be limited to two extreme points, ‘a’ and ‘b’. The area under the curve shown in Figure 2 is calculated using Trapezoidal rule. The area value with unique frame (sample) number becomes the single point of the curve shown in Figure 3.

Number of frames in 1 s = 20.

Number of frames in 1 min = 1200.

Total number of frames in 5 min = 5 × 1200 = 6000.

These 6000 values of the area cover under the curve of each radar sample data relate to respiration, heartbeat and noise that includes eyeblink, eyeballs movement and other ambient movements that present in the effective range of radar for the whole 5 min duration and stored in a vector A as shown in Figure 3.

The frequency spectrum of vector A is obtained by taking Fourier transform as shown in Figure 4. The frequency of signal acquired from UWB lies between 0 and 10 Hz, as shown in Figure 4.

The maximum frequency of respiration rate of an adult is 0.4 Hz [55,56,57,58]. A filter with a cut-off frequency of 0.4 Hz is required to acquire the respiration rate. In our case, when we take the normalized frequency, the cut-off frequency 0.4 Hz maps to 0.04.

Thus, a respiration signal is extracted by applying a tenth order low-pass Butterworth filter with a cut-off frequency 0.04 to the normalized spectrogram of the radar data frequency shown in Figure 5, removing the higher frequency noise with the outcome shown in Figure 6.

Finding the number of peaks and subsequently dividing this with number of durations in minutes provides a respiration rate per minute (RPM).

The RPM derived from the UWB-radar-acquired chest movement was validated using a commercially available pulse oximeter device, as shown in Figure 7. Three individuals were chosen for the validation experiment that includes one male and two females between the ages of 25 and 30 years. The individuals were instructed to sit comfortably on a chair facing the radar, with the pulse oximeter attached to the index finger of their left hand. Each subject’s chest movement was recorded twice, both times for 1 min. Table 1 shows the results validating the Extracted RPM using the proposed technique with the commercially available pulse oximeter. Table 1-column-3 values are taken from pulse oximeter and are merely used as a gold standard to compare with. Table 1-column-4: the RPM values extracted by the proposed algorithm using UWB.

Forty professional male drivers commonly carrying out long intercity driving hours (10 h approx) were selected for this experiment. For non-drowsy data collection, chest movement was acquired before driving in late evening, and for the drowsy data, the same subject’s respiration data were acquired immediately after they finished their arrival from a 10 h driving shift. During the data collecting procedure, drivers were encouraged to position themselves in front of the radar. To ensure that drivers were assessed as soon as they arrived from their journey, a test bed was set up in an empty room at Manthar Transport Company’s terminal, Sadiq Abad, Punjab, Pakistan. Ethical approval for this work was sought and approved by the Khwaja Fareed University of Engineering and Information Technology. Each participant was presented with the investigation, the data collection process, the experiments to be carried out and provided with a consent form that was read through with them. Participants were requested to sign the consent form to take part in the experiments. A minimum of 1 m distance between the radar and the subject at the subject’s chest level was maintained, as shown in Figure 8. The radar can cover 9.4 m distance from an object to the radar transmitter–receiver point. Within this range, any movement can be captured. The 1 m distance is chosen on the assumption that driver could be anywhere within this range while driving. With the system placed in or around chest level, the subject should be in front of the effective radar range; in practical implementation, it could be 0.2 m–0.5 m (distance from dashboard to human body).

The chest movement of each participant was recorded for 5 m using the radar before and after 10 h drive and RPM (average respiration rate of 5 m) is extracted by the proposed method. The RPM, along with the subject’s age and labels, i.e., non-drowsy and drowsy, was stored in a CSV file. Figure 9 and Figure 10 show raw respiration signals from the first minute in drowsy and non-drowsy states.

Driver movements do not affect the performance of the proposed system. To ensure the above fact, an experiment was performed in a real driving environment; participants were asked to drive the car at various speeds, the Appendix A can be seen in the Appendix A.

A pulse oximeter (encircled blue) was attached to the subject’s left index finger while driving, and radar (encircled red) was mounted on the dashboard, as shown in Figure 11. Six male subjects participated in this experiment, and data is recorded from pulse oximeter and radar parallelly while driving in real environment. The respiration rate calculated from chest movement recorded during driving is validated using a pulse oximeter, and the findings are given in Table 2.

It is evident from Table 2 that the RPM difference between pulse oximeter and radar is 0.58. This shows, the driver moment is not largely affected, since every periodic movement has a certain frequency, e.g., blinking eyes, eyeball movement, heart rate, respiration rate, even the posture or body movement do not affect the data collection as they occur at frequency bands outside the spectrogram frequency of interest.

In order to investigate automated drowsiness detection and optimize outcomes, various machine learning models suited for these types of data sets were implemented and compared.

### 3.1. Support Vector Machine (SVM)

SVM is a supervised ML model that uses kernel tricks to transform dataset. An optimal boundary is set between outputs based on these transformations. This model can be used for both regression and classification [59]. Three main parameters used in this manuscript to tune SVM are given in Table 3.

### 3.2. Decision Tree (DT)

DT is a supervised ML model used in classification problems that can accommodate numerical and categorical data, building models in the form of trees. This model divides the dataset into small subsets, and a decision tree is incrementally developed. Finally, a tree with decision and leaf nodes is formed. Decision nodes can have two or more than two branches, while leaf nodes represent a decision/prediction [60]. The parameters that are used to tune the classifier are given in Table 3.

### 3.3. Extra Tree Classifier (ETC)

Extra tree classifier is an ensemble ML model. It creates a large number of DTs from training data. Predictions are made by majority voting in classification and averaging the prediction in regression data. ETC randomly samples the features at split point of each decision tree [61]. The tuning parameters of ETC used in this manuscript are given in Table 3.

### 3.4. Gradient Boosting Machine (GBM)

A method of converting weak classifiers into strong classifiers is called boosting. In boosting, a new tree is trained on the modified version of original data. GBM makes decision trees in an additive, sequential and gradual manner. It finds the shortcomings of weak decision trees based on the gradients in the loss function as shown in equation.
y=ax+b+e
where ‘e’ is the error term in this equation.

The Loss function is the difference between an actual and predicted value and indicates how good a model is performing on a given dataset [62]. Parameters that are used to tune the GBM classifier are given in Table 3.

### 3.5. Logistic Regression (LR)

LR is an ML model based on probability. It is also known as linear regression but uses a complex cost function called Sigmoid function. LR limits its cost function between zero and one. Sigmoid function maps any real value between 0 and 1. Mathematical representation of Sigmoid function is given in equation below. It gives an S-shaped curve.
$$g\left(x\right)=\frac{1}{1+e^{-x}}$$

LR returns output class based on the probability when an input is passed through the prediction [63]. Three main parameters used to tune the LR are given in Table 3.

### 3.6. Multilayer Perceptron (MLP)

MLP is a fully connected multilayer neural network that is composed of generally three layers, namely, input layer, hidden layer and output layer. Hidden layers could be more than one, known as deep neural network. The input layer is the lowest layer that receives data from a data set. The neural network is drawn with one neuron per input value or column in a dataset as an input layer. Hidden layers receive a set of weighted inputs and generate output through an activation function. Output layer is responsible for generating a value or vector of values that matches the format required by problem.

Different parameters were used to tune the ML models and parameters where the models were generalized are tabulated below.

## 4. Results

A structured dataset was formed with RPM (average respiration rate) and age of subjects (drivers) along with labels (non-drowsy, drowsy). The data set was divided into training and testing sets in 70% and 30% ratios, respectively. The test data set that was kept unknown to the model was set aside for evaluating the classifier. Different supervised ML algorithms including SVM, DT, ETC, GBM, LR and MLP were used for classification of data.

A 15-fold cross-validation was used for training to tune the hyperparameters of the classifiers. The non-standardized feature vector comprising RPM (average respiration rate of 5 m) and age were fed as input to the ML models to classify into labels (drowsy or non-drowsy). Subsequently, ML models were evaluated on previously split test data. The accuracy of ML models that is shown in Table 4 was computed from the predictions made on test data by using the following formula.
$$\mathrm{Accuracy}=\frac{\mathrm{TP}+\mathrm{TN}}{\mathrm{TP}+\mathrm{TN}+\mathrm{FP}+\mathrm{FN}}\times 100$$
where

TP = the subjects correctly identified as drowsy while subjects were actually in drowsy state.

TN = the subjects correctly identified as non-drowsy while subjects were actually in non-drowsy state.

FP = the number of subjects incorrectly identified as drowsy while subjects were actually in non-drowsy state.

FN = the number of subjects incorrectly identified as non-drowsy while subjects were actually in drowsy state.

The findings in Table 4 reveal that SVM outperformed other ML models, indicating that SVM is a more generalized approach. Tree-based ensemble learning methods like ETC are mostly used for multiclass classification where the dataset comprises non-linear and categorical data [64]. SVM performs better in binary classification where the dataset has a linear dependency creating a hyperplane dependent on the data points that best separate the labels. A tendency towards linear dependency in Figure 12 can be seen between age and respiration per minute. Figure 12 comprises 80 points (40 for each label of drowsy/non-drowsy); however, some of the points overlapped, which is why Figure 12 has fewer points. The result reflects the fact that respiration per minute decreases in a drowsy state [26]. Further, in a drowsy state, the decrease in RPM is higher in younger as compared to older subjects. The same pattern follows in non-drowsy state with a bit shift toward right on the *x*-axis.

A threshold of 18.5 is compared with other classifiers used in this study at the same test data. Findings in Table 4 shows that SVM and threshold show same accuracy, but SVM shows better precision, recall and F1 score, which means SVM is better generalized.

## 5. Discussion

As mentioned earlier in the manuscript, driver drowsiness is one of the major causes of accidents worldwide. Many approaches that have been adopted to detect driver drowsiness via physiological signals employ invasive or wearable sensors. Invasive sensors can be used in virtual or controlled environments, but in real-world circumstances, they require driver commitment and compliance as well as the possibility of driver discomfort and privacy issues. A non-invasive, camera-free method is presented to detect drowsiness based on respiration rate acquired from the chest movement recorded by UWB radar. The accuracy achieved by the proposed method is compared with some state-of-the-art techniques mentioned by the investigators to detect drowsiness. From Table 5, it can be seen that [37,43,44] show better accuracy than the proposed method. However, [37] used an oximeter, and in [43], thermal imaging was used to acquire the respiration rate. In [44], researchers used a Nexus-10 device. Apart from invasiveness characteristics, the pulse oximeter must be connected correctly to the body in order to obtain the breathing rate, whereas the Nexus-10 device acquires physiological signals by various on-body sensors. These two devices are less likely to have a meaningful impact in a real-world driving scenario since they must have driver compliance and commitment and rely on placement-correction software being used. Additionally, the pulse oximeter used by [37] gives an unreliable reading when the fingers of the subject are not dry or contaminated with oil, grease and dust, which is common during driving [65,66]. In paper [43], the team used thermal imaging to estimate respiration rates from changes in temperature below the nostrils during inhalation and exhalation. This is unlikely to succeed in the real environment due to the likelihood of sudden head movement during driving. The proposed method uses UWB radar to acquire respiration rates from chest movements UWB radar is a non-invasive method that can acquire chest movements, in this case, over a range of 0.2–1.6 m. Other driver movements such as blinking, head movements, etc. can be identified as subject to appropriate filtering and data logging, but importantly do not affect respiration rate collection. The proposed method has limitations which the investigators will now build on following this trial: initial results are based on a small data set; more data is required to enhance the accuracy of classifiers. During the data-gathering procedure, drivers of a certain age (30–50 years) and same ethnic background were considered; the team will increase diversity for further tests.

## 6. Conclusions

Driver drowsiness is one of the leading causes of accidents. Many investigators have presented systems to detect drowsiness using respiration rates acquired by invasive sensors. A non-invasive, i.e., non-wearable, non-camera-based driver drowsiness detection system based on wirelessly extracting respiration rates is designed and presented. The respiration monitoring system was validated with a readily available, off-the-shelf, medically accepted, commercial pulse oximeter device. After the validation of the respiration rate monitoring system, it was used to acquire the respiration rates of drivers in pre- and post-driving states. A structured dataset was formed based on the respiration per minute (RPM), age and labels. Machine learning models were trained and validated on the dataset. The ML algorithm SVM shows better accuracy, outperforming other ML models trained in this manuscript. This research provides ground truth for the verification and assessment of UWB as an effective technology for driver drowsiness detection based on respiration. In the future, data will be collected in a real driving environment, or the control of a vehicle can be transferred to a computer for the detection of drowsiness.

## Figures and Tables

**Figure 1 sensors-21-04833-f001:**
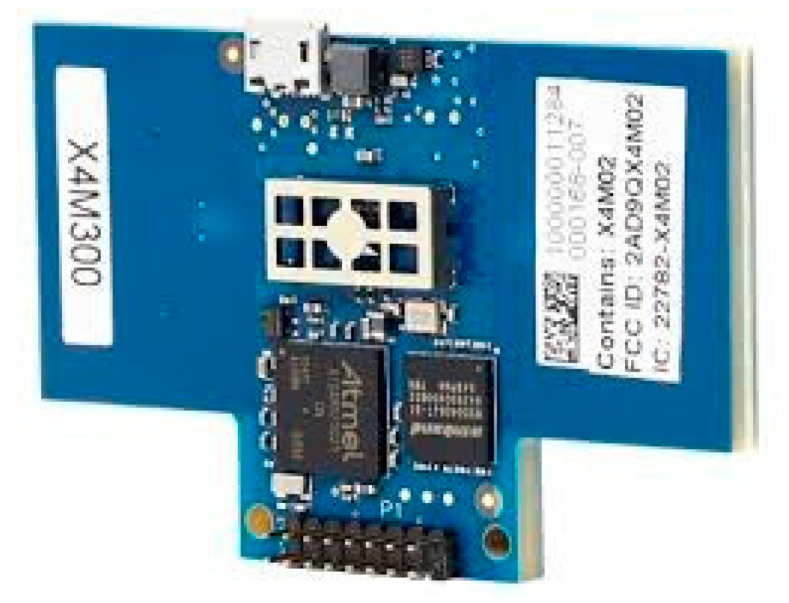
X4m300 UWB radar.

**Figure 2 sensors-21-04833-f002:**
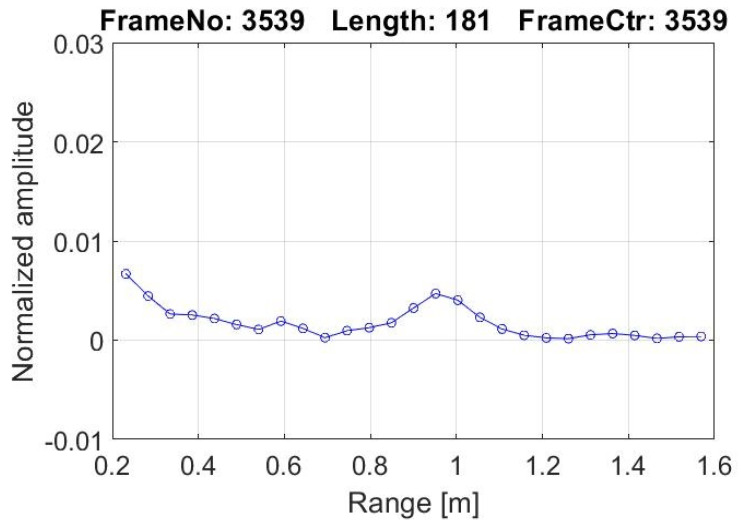
Plot of signal during data collection.

**Figure 3 sensors-21-04833-f003:**
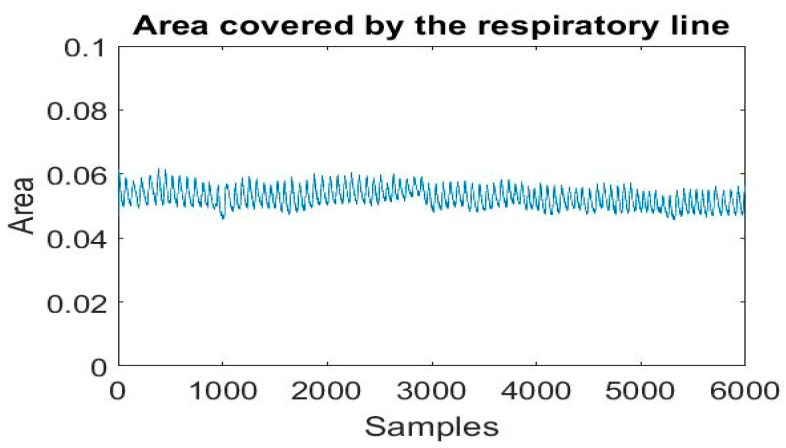
Noisy signal after finding area under curve.

**Figure 4 sensors-21-04833-f004:**
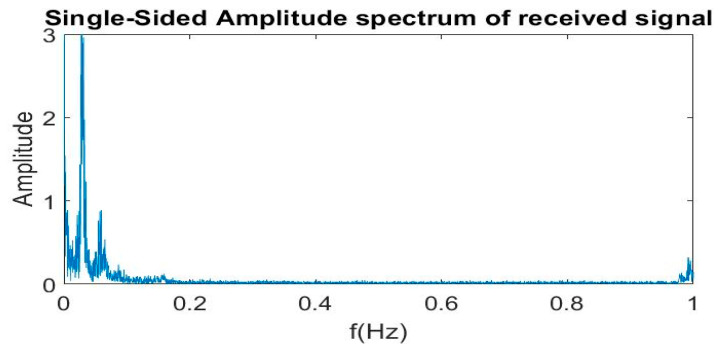
Fourier transform of noisy signal.

**Figure 5 sensors-21-04833-f005:**
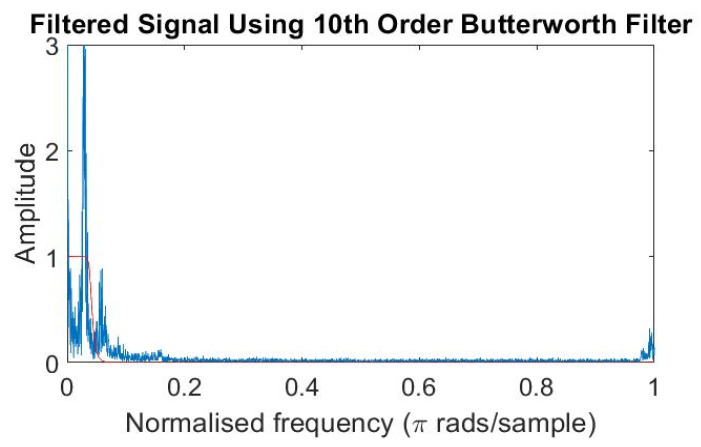
Tenth order low-pass Butterworth on a noisy signal.

**Figure 6 sensors-21-04833-f006:**
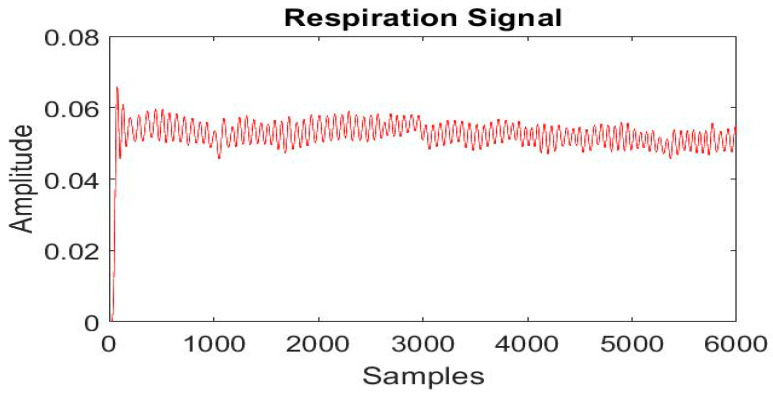
Respiration signal.

**Figure 7 sensors-21-04833-f007:**
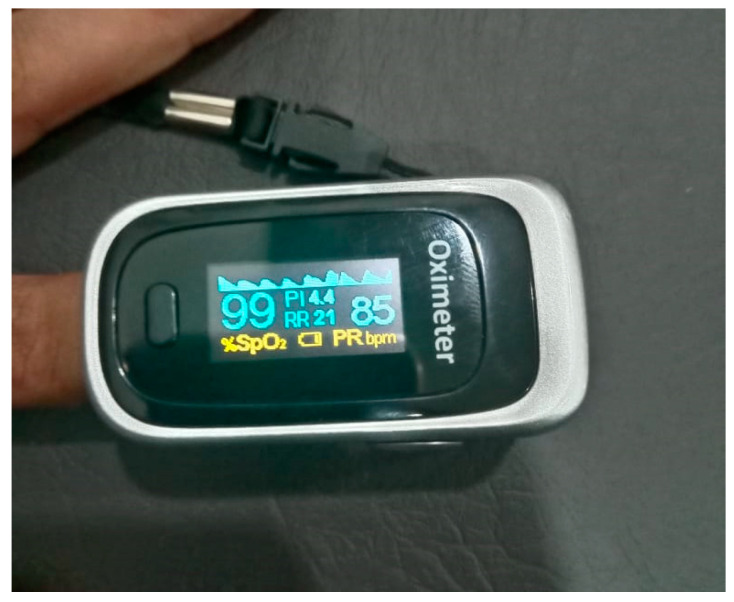
Pulse oximeter.

**Figure 8 sensors-21-04833-f008:**
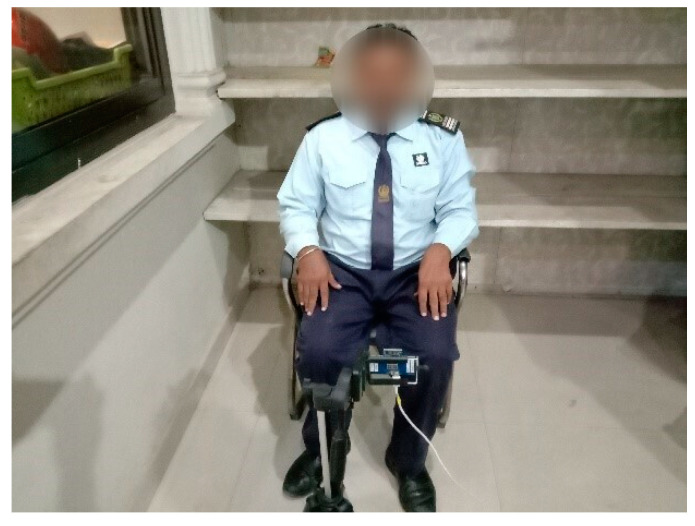
Subject sitting in front of the radar.

**Figure 9 sensors-21-04833-f009:**
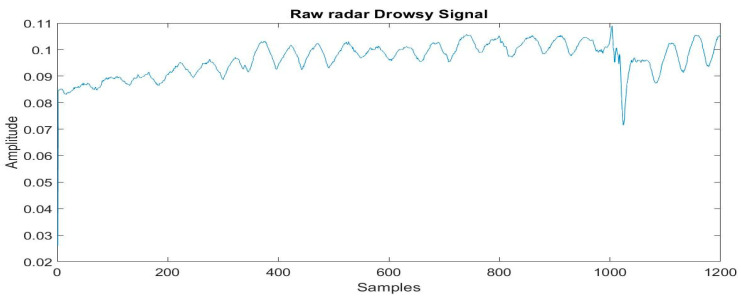
Radar signal for drowsiness.

**Figure 10 sensors-21-04833-f010:**
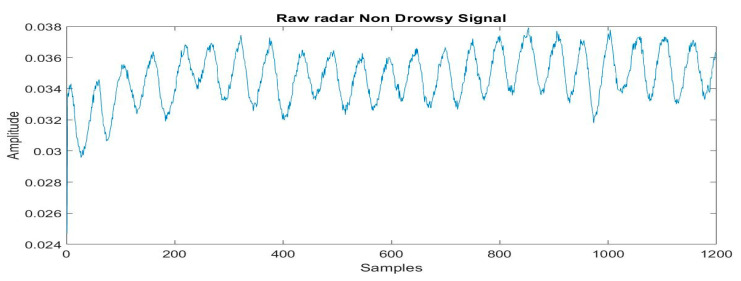
Radar signal for non-drowsiness.

**Figure 11 sensors-21-04833-f011:**
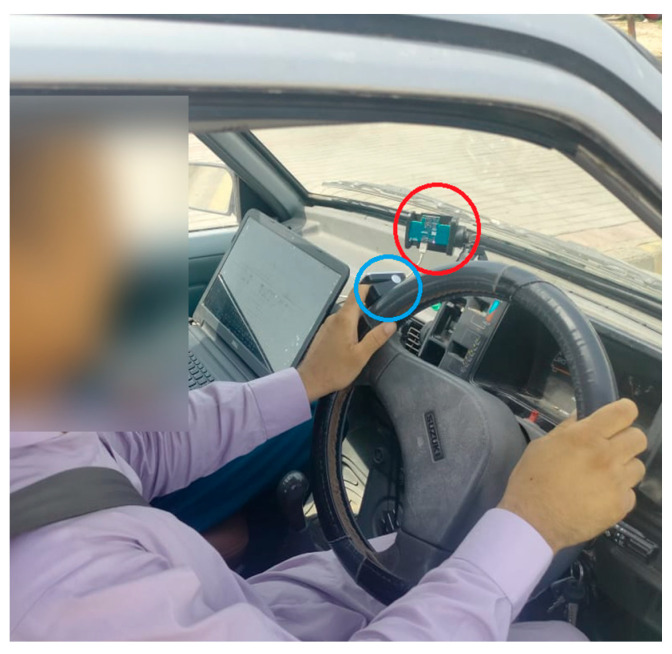
Subject driving car.

**Figure 12 sensors-21-04833-f012:**
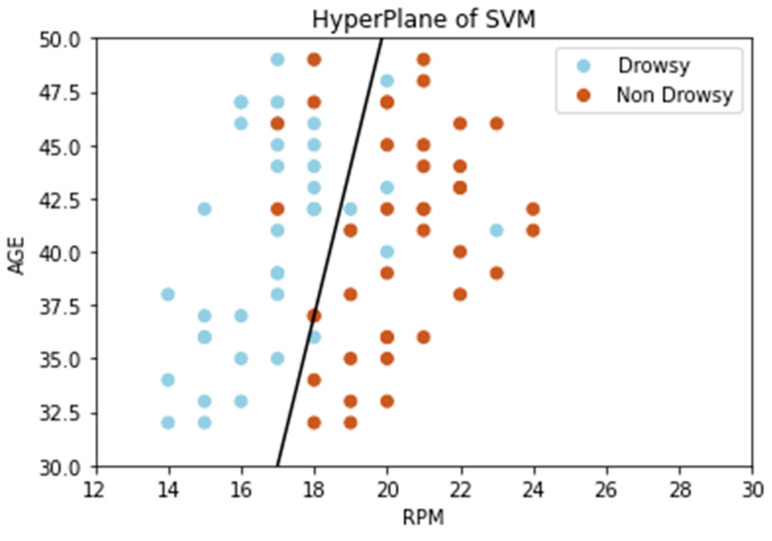
Hyperplane draws by SVM after expansion of dataset.

**Table 1 sensors-21-04833-t001:** Validation results.

Subject	Test Time	Respiration Acquired by Pulse OXIMETER	Respiration Acquired by Proposed IR-UWB Method
**Subject 1**	09:57–09:58	15	15
	11:03–11:04	18	18
**Subject 2**	10:05–10:06	16	16
	11:09–11:10	17	17
**Subject 3**	10:11–10:12	21	21
	11:17–11:18	12	12

**Table 2 sensors-21-04833-t002:** Results of experiment conducted in real environment.

Subject	Respiration Rate Acquired by Pulse Oximeter	Respiration Rate Acquired from Chest Movement by Proposed Method	Car Speed in kms/Hour
**Subject_1**	15	16	20
	17	17	40
**Subject_2**	19	18	20
	17	18	40
**Subject_3**	16	17	60
	15	15	45
**Subject_4**	17	18	20
	16	16	30
**Subject_5**	19	18	60
	19	20	50
**Subject_6**	17	17	50
	19	19	20

**Table 3 sensors-21-04833-t003:** Parameters used to tune classifiers.

Classifier	Values of Parameters Used during Training in This System
SVM	Kernel = ‘linear’, c = 1.0, gamma = ‘scale’, degree = 3
DT	Criterion = ‘gini’, splitter = best, maximum depth of tree = none, minimum number of samples = 2, minimum required leaf nodes = 1, random states = none, maximum leaf nodes = none, minimum impurity decrease = 0.0
ETC	Number of estimators/trees = 100, criterion = entropy, minimum number of samples = 2, maximum number of features to consider during classification = auto
GBM	Loss = deviance, number of estimators = 100, criterion = friedman_mse, minimum number of samples = 2, minimum samples to be a leaf node = 1, maximum depth = 5
LR	Penalty = L2 regularization (ridge regression), solver = liblinear, maximum iteration = 100
MLP	Hidden layers = 2, neurons = 100 for each layer, epochs = 700, activation = ‘relu’, loss_function = ‘stochastic gradient’, solver = ‘adam’

**Table 4 sensors-21-04833-t004:** Accuracies of classifiers.

Classifier	Accuracy	Precision	Recall	F1 Score
SVM	87%	0.86	0.88	0.86
LR	70%	0.68	0.69	0.68
GBM	62%	0.59	0.59	0.59
ETC	70%	0.68	0.69	0.68
DT	62%	0.59	0.59	0.59
MLP	70%	0.68	0.69	0.68
Threshold 18.5	87%	0.73	0.75	0.73

**Table 5 sensors-21-04833-t005:** Comparison with different studies.

Reference	Accuracy
[37]	100%
[43]	90%
[44]	93%
[46]	82.55%
Proposed method	86%

## Data Availability

The dataset is in use in another research and will be made available, once the other research is done.

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
