# Peer review of "Non-Invasive Driver Drowsiness Detection System"

_sensors, 2021, doi:10.3390/s21144833_

Round 1
Reviewer 1 Report
The authors do a commendable job on tackling the important issue of drowsiness and to explore a method for potentially monitoring driver condition. A strength of the paper is the very clear and informative use of various machine learning techniques for the analysis of data. In particular, listing the various parameters for each of the methods allows for reproducibility by other research teams. The comparison of results illustrates the pros/cons of the various techniques and this is of service to the readers.
The description of current literature on drowsiness is complete, as is the technical description of the IR-UWB system; such detail is helpful to others.
The premise of drowsiness exclusively to respiratory is present , but for this to be clinically relevant, then either there need to more than a few citations (other than self-citing) in the open press for the credibility to be enhanced [Lines 71-74]. With 5 subjects the focus of this study the researchers have done a service in building a platform for further research more-so than actually affirming the hypothesis. IN particular, the need to use machine learning in the first place is possibly because a significant clinical effect (drowsiness) is being teased out of a small number of subjects and with only one parameter. If this can be validated further, then the research team will have accomplished a significant contribution to the field. As is, the platform is commendable in its clarity and the way the authors describe its usage.
The idea that all other monitoring methods are "invasive" and that a radar-based approach is not is a challenging statement to accept. If the more conventional definition of "invasive" includes "percutaneous" than both EEC and radar are non-invasive. The nature of the measurement described as "off-body" and convenient would not cause the reader to pause, particularly those with a clinical background.
Lines 80-153 provide an abundance of information, and it is suggested that the presentation be modified with some sub-heading to allow the reader to follow the various technology modes easier. And it is disorienting that Lines 154-155 are actually critical in that the authors are now transitioning to their research effort. The transition catches the reader of guard and you have to re-read; that is, the reader is expecting some sort of summary statement, but instead a direct transition to the current methods and materials.
The researchers make the point of the SVM outperforming other methods which is VERY important since often ML researchers prefer to find the most complicated numerical method, and it is refreshing that a well-proven methodology has application.
The reviewer pushed back a bit on Lines 322-323 in the results being the "ground truth" but rather the work reflects the development of a commendable platform for further work with a much larger cohort.
Author Response
We would like to thank the reviewer for his/her thoughtful comments and efforts towards improving the manuscript. In the following, we highlight general concerns of reviewer that were common and our effort to address these concerns.
We have splitted the introduction into introduction and related work section and added a conclusion. Improve the discussion section by adding bit more technical detail.

Reviewer 2 Report
This paper presents a method to detect drowsy driver states based on respiration rate detection using a impulsive radio ultra-wide band (IR_UWB).
I have some comments and questions:
-
Paragragh Line 53: you could add refs for each physiological approach.
-
Explain in the introduction (or discussion) section, the originality and advantages (you have explained some) versus the limitations of your approach. Are you the first to use IR_UWB to study respiratory rate and drowsiness? Can driver movements affect the performance of your method?....
-
Add a paragraph in the introduction to explain the different sections of your article.
-
The results of table 1 catch my attention. Is there a logical explanation that the results are identical for both techniques? Are these results the same regardless of the distance and orientation of the sensor? If the results are the same, what would be the advantage of using your method instead of an oximeter? For example, what would be the advantage of using your sensor instead of https://doi.org/10.3390/electronics8080890?
-
Could you please provide a figure that shows a radar signal for drowsiness versus no drowsiness?
-
From line 188: you try to filter some artifacts using a lowpas filter. However, you do not show any significant artifact. Why a cut off frequency of 0.004? How have you selected 0.004?
-
Line 224-225: Does it mean that the sensor has to change according to the position of the driver's chest and consequently there is no standard position for everyone?
-
Paragraph 287:I don't think it is necessary to explain the concepts of overfitting and underfitting.
-
Results of table 7 and 8 catch also my attention: I don't understand why you analyze 27 participants and then 40 .The normal thing is to carry out a single analysis with the maximum number of participants available. In any case, I cannot explain that ETC does not have overtiffting with 27 participants and this overfitting appears with 40 participants.
-
What is the feature vector for each participant [RPM] alone, [age, RPM]....?
-
From Fig. 9 you could select a threshold of 18.5, right?. If possible, what advantage would SVM or another method provide, with respect to this threshold?
-
Line 320: again, I don't see why it seems interesting to compare small datasets with larger datasets ... In my opinion, this comparison is not necessary.
-
In this article, there is no quantitative analysis regarding other solutions . Although the data and methodology are different, the classification results of other technologies could be compared with the results of this article.
Author Response

(The authors gave the same response as above.)

Reviewer 3 Report
This work presents an interesting approach to address a current and relevant problem. However, in order for the study to have sufficient relevance to the research field, it must first improve two main aspects:
First, the current introduction should be divided into two sections: introduction itself and related works. The analysis of the past works should be improved, focusing on gathering useful information that allows a clear comparison with the results obtained in the study and the difference in the approaches regarding the solution proposed by the authors.
Second, and more importantly, the machine learning models used, although relevant depending on the problem to be addressed, are all belonging to the branch of classical ML algorithms. Since the results of effectiveness obtained in the classification are not competitive with respect to the most current ML algorithms, I consider essential the evaluation with some of these “newer” models. I suggest training a Fully Connected Layer Neural Network model with one and two hidden layers at least. The complexity of these models is not enough to imply a delay in the execution times, and the effectiveness of these models has currently been shown to substantially improve systems effectiveness that address problems such as the one presented by the authors in this manuscript.
Additionally, the discussion is sparse and does not go deep enough to provide significant value to the research area. Authors should strive to draw more elaborate conclusions that reinforce the importance of the study conducted.
Author Response

(The authors gave the same response as above.)

Reviewer 4 Report
The paper proposes a non-invasive system to recognize a drowsy or non-drowsy state in drivers. It uses an impulsive ratio ultra-wide band (IR-UWB) radar that allows the analysis of the respiration rate of the subjects. Various Machine Learning (ML) techniques are used for binary classification (drowsy state vs non-drowsy state) and a final comparison performance of these algorithms is then presented.
Although the topic is interesting and the use of the IR-UWB radar for psychological signal acquisitions can be helpful in various applications, I am sorry to say that in my opinion the presentation of the highlighted contributions and the quality of the manuscript, as in this current version, are not suitable for publication. Moreover, the results (even if they appear to be somehow promising) seem to be too preliminary and need further investigation, in particular regarding the possible artifacts, due to motion, posture changes, etc., which are expected in a real scenario.
Other issues are reported as follows:
- The Introduction section could be shortened, and the innovative contributions of the proposed system, also when compared to other state-of-the-art works (e.g., [23]), should be better evidenced.
- Section 2 “Materials and Methods” should be carefully revised. Figure 1 showing the radar, as described in the text, appears wrongly as “Figure 2” looking at the caption (the caption of Figure 1 reports “Figure 2”), the same happens for Figure 2 (“Figure 1” appears in the caption of Figure 2). How the data are collected from radar (including the figure that shows these data), and the procedure for the derivation of the respiration signal, should be explained in much greater detail and more precisely (e.g., the exact definition of area under the curve and its use should be provided). In the text it is reported that “three subjects took part in the validation process” but the results of six subjects appear instead in Table 1. The choices made on the parameters that characterize the different ML techniques should be justified (see also the next comments regarding the validation set).
- The experimental setup should be better detailed. How were the people asked to behave during the test? What about the 10 hours of driving, between the two different phases of the test? Did the subjects drive in a city environment? Please add more details about the setup of the experiment.
- The result section should be better described. It is not clear why one should initially only test 27 subjects and then add the others, in order to reach the final number of subjects (40). The fact that only the RPM is used for each one of the five minute intervals recorded from each subject is not described clearly: it seems that only a sentence in the abstract states it. However, is the average of the RPM values, computed from each five minute interval, used as input to the various ML algorithms, along with the age? Or are the five RPM values (one for each minute) used? Please clarify this point. Why do you choose to divide the data in training and test set only, without considering the validation set? This could be used to fine-tune the model hyperparameters of the various ML algorithms. Given the limited amount of data, a cross-validation procedure could be employed in this case. Please discuss this better. In summary, the overall ML setup is not described appropriately (it seems that the age of the subjects is taken into account, but this is not previously discussed).
- The Discussion section should be expanded, and a Conclusion section should also be added to the manuscript.
- As already stated, the quality of the manuscript needs to be improved. The literary style needs a revision to correct the wording and several typos (e.g., “these concepts doesn’t” -> “these concepts do not”, “SVM perform” -> “SVM performs”, “leaf nodes represents” -> “leaf nodes represent” etc.). Sometimes it is not easy to completely understand the meaning of all sentences. As minor comments, I noticed that it also happens that a blank space is often present between a word and the final period (e.g., “frame size [37] .” -> “frame size [37].”) or that a blank space is missing (“3.5 seconds[39].” -> “3.5 seconds [39].”), etc.
Author Response

(The authors gave the same response as above.)

Round 2
Reviewer 2 Report
Ok about your answers. Only two considerations: 1) I do not agree with the explanation of question 6. It is true that an oximeter can be considered invasive but the article describes that the sensor is located on the steering wheel. I think it is a reference that you could comment on in the introduction. 2) in relation to question 13. I was referring more to if you could add what is the precision of the threshold 18.5 to compare it with the other methods. The result is not better, but I think it would be good to quantify how worse this threshold is relative to the chosen methods.
Reviewer 3 Report
The authors have strived to address the concerns raised in the first round of review. The main problem I find is the writing of the article. Several sentences, paragraphs and sections are disjointed, some are not understood correctly, and there are several incorrect or repetitive terms.
Authors should also correct the following:
- The accuracy metric is not formally presented in terms of what is considered a positive and a negative sample in this analyzed classification problem.
- Line 292: “three hidden layers” is incorrect. Both first (Input) layer and last (output) layers are not considered hidden layers.
- Lines 295-296: “There is a single neuron in the hidden layer of the network structure that explicitly outputs the value.” Again it is incorrect, referring to what was explained for line 292, and also because an output layer can have several nodes and with it several output values. In the case of the problem addressed, which is binary, it is true that there is only one output.
Reviewer 4 Report
The paper has been modified trying to answer the reviewers’ comments. Even with these changes, acknowledging that the authors made an effort to improve the work, the paper in my opinion still needs some further clarifications in order to make it suitable for publication.
As already stated in my previous review, the results are somehow promising, but seem only preliminary, and more details about the possible artifacts expected in a real scenario should be included in the manuscript. The authors responded on that saying that driver movements do not affect the performance of the proposed system. This has to be confirmed by actual data in a real driving scenario, or at least a reference where that has been demonstrated should be added. A simple experiment, for example where one subject is sitting on a chair and moving the body, or mimicking typical car shaking could be added.
The procedure that enables the collection of the data from the radar has been slightly expanded, but its explanation should be improved since it is still not clear enough. It is difficult to understand all of the details in the first reading. As an example, the statement about the 27 points recorded in a window of 9.9 meters seems to be in contrast with Figure 2, where one can count around 16/17 points in the first 1.5 meters already. Please explain this better.
The results still need to be better described. Are the feature vectors standardized? It could be more clearly indicated that only training and cross validation on data have been carried out. Is the cross validation phase only used for hyperparameter tuning? How is the accuracy finally computed? The test on unseen data is missing, and this is another detail that could be clearly written. This phase should be included in the authors’ future experimental results. More details should also be added on the text regarding Figure 11. One would expect to find 80 points shown in this figure (40 subjects for each drowsy/non-drowsy class) but less points appear on that figure. Please clarify this better.
In addition, the comparison between the results obtained in the manuscript and the others reported in the cited papers should be better discussed. For instance, it seems that the only difference between the proposed scheme and [37] is the fact that [37] uses the oximeter. But if this is true, since the authors verify that they obtain the same RPM values recorded by the oximeter at the same time, the scheme in [37] appears to perform better than the one proposed here anyway. Please justify this.
The quality of the manuscript, from the English language point of view, still needs to be improved. There are some typos in this version of the manuscript as well, e.g., “[37][43, 44] shows” -> “[37][43, 44] show”, “propose method” -> “proposed method” etc.).
